# Nursing supervisors' perspectives on student preparedness before clinical placements- a focus group study

**Ann-Chatrin L. Leonardsen**[1,2]*, **Siri E. Brynhildsen**[1], **Mette T. Hansen**[1], **Vigdis A. Grøndahl**[1]

1 Department of Health and Welfare, Østfold University College, Halden, Viken, Norway, 2 Department of Anesthesia, Østfold Hospital Trust, Grålum, Viken, Norway

* ann.c.leonardsen@hiof.no

## Abstract

Clinical placements and supervision is an integral part of nursing education internationally. There are significant differences between students' expectations of clinical learning and their fulfillment. Few studies have focused on supervisors' perspectives on clinical placements. The objective of this study was to explore nursing supervisors' perspectives on students' preparedness for clinical placements.

### Methods

The study was conducted in a county in Southeastern-Norway, with 317.000 inhabitants, and within one hospital and one university college catchment area. Focus group interviews were conducted in the periode August to December 2018. Data were analyzed using Hsieh and Shannon's conventional content analysis.

### Results

34 nursing supervisors participated, three intellectual disability nurses and 31 registered nurses, working in four different primary healthcare wards and four different hospital wards. Participants' age ranged from 23 to 58 years, one male only. Through the analysis we derived the category 'Shared responsibility for preparation' with subcategories a) Individual initiative, and b) University college facilitation.

### Conclusions

Findings indicate that there is a gap between nursing supervisors' expectations and reality regarding students' preparedness for clinical placements. Moreover, nursing supervisors did not seem to focus on their own role in student preparedness.

**Data Availability Statement:** Data cannot be shared publicly due to confidentiality issues, and the possibility that individuals may be recognized in the data. Data may be shared upon reasonable

request to a data contact person: Ann Karin Helgesen, Professor and Head of Research at Østfold University College. Email: ann.k. helgesen@hiof.no.

**Funding:** ACL received fundings from Østfold University College (https://www.hiof.no/). The funders had no role in study design, data collection and analysis, decision to publish, or preparation of the manuscript.

**Competing interests:** The authors have declared that no competing interests exist.

## Introduction

Nurse education in Europe is characterized by different structures, standards and approaches to the relationship between theoretical and practice based learning [1, 2]. The European Commission has, through the Bologna Treaty Process, promoted greater harmonisation of educational systems in the European area [3]. Such harmonisation seeks to increase the employment and educational mobility of nursing staff and students.

Clinical environments offer registered nursing- (RN) and intellectual disability nursing (IDN) students a possibility to gain clinical training, that requires them to employ their knowledge and skills to develop qualifications to be able to take care of patients [4, 5]. To provide high quality care, clinical supervision is widely used as a professional process to support undergraduate nursing students [6], in the development of professional competence and confidence [7]. The support provided through clinical supervision helps the students to link theoretical knowledge provided through lectures, with patient care in the clinical environment [6, 8, 9]. Clinical placements enables supervisors to directly observe students in the clinical environment, and also gives an opportunity to build a supportive relationship between the supervisor and the student [10]. Such supportive relationship will hereby optimize students' learning [11]. Lack of this supportive supervisor–supervisee relationship has been found to result in negative clinical learning experiences [12]. Scholars internationally emphasize the need to strengthen clinical supervision, in order to contribute to nursing student's personal development and increased competence [8, 11].

Expectations may be understood as reasonable likelihood that something will occur [13]. Expectations are linked with motivational (interests) and cognitive (assessment) aspects of behavior, and have great impact in the processes of decision-making, as well as in academic and workplace performance [14–16]. Statistically significant differences have for example been found between the nursing students' initial expectations and their fulfillment at the end of the academic year for all the factors and in all years of the nursing bachelor degree program [17]. Clinical placements are a central part of the nursing education cirrucula. In Norway, nursing services are provided by either registered nurses (RNs) or intellectual disability nurses (IDNs). Both cirrucula includes three years full-time studies, and results in a total of 180 ECTs (European Credit Transfer and Accumulation System). Clinical placements and supervisors are essential in the RN and IDN students' path to being a competent professional. Educators expect students to be actively engaged in contributing to their clinical placement experience and determining appropriate learning outcomes. In the clinical context, this requires that educators collaborate with students' to identify their learning needs and ensure they are provided with opportunities together with the clinical supervisor [12, 14]. For the student, this requires that they are well prepared in terms of basicknowledge and attitudes during their clinical placement periods. This include that they accept a share of the responsibility for planning and preparing for the learning experience [18]. Research on supervisors' perspectives on clinical placements, as the ones who direct the students to gain professionally related skills in the clinical environment, is limited [18, 19]. Moreover research on supervisors' expectations, and their perspectives on students' preparedness are lacking. Consequently, the aim of this study was to explore RN- and IDN supervisors' perspectives on students' preparedness to clinical placements.

## Materials and methods

### Design

The study is in-line with the Consolidated criteria for reporting qualitative research–COREQ [20]. The study had a qualitative design, utilizing focus group interviews to explore RN and

IDN supervisors' perspectives on students' preparedness for clinical placements. A qualitative approach is appropriate when aiming to explore how individuals experience a phenomenon, dependent on their background, interests and interpretation [21]. Focus groups are appropriate when exploring experiences and perceptions, enabling further development of these through group discussions among participants. During focus group interviews interaction, dialogue, and shared reflections between the participants, can increase and deepen the understanding of the phenomenon that is explored [22, 23].

## Setting

The study was conducted in a county in south-eastern Norway concisting of 18 municipalities, and within the cathcment area of one hospital trust that covers 317 000 inhabitants. Within the municipalities there are five decentralized acute care wards and five casualties. Moreover, there are 34 nursing homes, and more than 4o home based nursing wards, in addition to institutional services for persons with intellectual and physical disabilities. The county adhere to one university college, which provides bachelor degree education for RN and IDN students. A total of approximately 1700 clincial placements are carried out every year, and approximately 160 registered nurses (RN) and between 45 and 95 intellectual disability nurses (IDN) finish their bachelor's degree from the university college each year.

## Sample

A purposive sampling strategy was utilized, where supervisors from all the different types of clinical placement wards were invited to participate. The researchers aimed at including supervisors from both primary healthcare services and different hospital wards. Pragmatically, the aim was to include participants to eight focus group interviews. Two medical and two surgical hospital wards respectively were randomly selected, as well as four wards in primary healthcare services. The RNs/IDNs nearest leader (RNs/IDNs themselves) in each ward selected participants assumed to be information rich, experienced with supervision of RN/IDN students. Inclusion criteria were; RN/IDNs who had supervised bachelor students within the last two years. No exclusion criteria were added.

The supervisors were given verbal and written information about the study and asked to participate by their manager. The supervisors who wanted to participate, signed an informed consent.

## Interview guide

A semi structured interview guide was developed by the authors, based on the respective bachelor program's curriculum, informal feedback from teachers at the university college and from supervisors, also including the researchers' own experiences and competencies from education and the clinical field (see S1 File). The interview guide was assessed and discussed in a research group, consisting of six RNs and one IDN with both clinical and educational experience (in addition to the authors), and no revisions were needed. The questions were used to guide the focus group interviews. Table 1 gives an overview of the interview guide.

## Procedure

The focus group interviews were conducted nearby the participants' workplace in an office or meeting-room away from the actual ward. The interviews started with an open dialogue where the participants were encouraged to freely discuss their experiences related to supervision of RN/IDN students during clinical placement. The interview guide was then used as a

**Table 1. Examples of questions from the interview guide.**

| | Probing questions | To include all participants |
|---|---|---|
| Could you please describe practical skills and your experiences of how the students learn these skills? | Could you please elaborate on that? | What about the rest of you? |
| Could you describe your expectations to the students preparedness for clinical placements? | Is it something students could have been better prepared for? | Please tell me more about that? |
| | Is it something most students are prepared for? | |
| How do you og by when aiming to learn students practical procedures in clinical placements? | Which procedures are important? | Please discuss, is this common for all? |
| | Why? | |

«checklist» to the ensure that the themes were covered. The focus group interviews were conducted by two and two participants of the research group, consisting of RN and IDN educators and clinicians, some with a PhD and some without (two males). Their roles were as active interviewer and observer respectively. All researchers were experienced RN/IDNs, with several years of professional practice, and/or several years of experience as educators. Moreover, a researcher with qualitative research experience participated in all interviews. The interviewers were not familiar to the participants.

The focus group interviews were conducted in the periode August to December 2018. The interviews lasted from 30 to 70 minutes (mean 50 minutes). The interviews were digitally recorded, and transcribed verbatim by an external transcriber, who had signed a non disclosure agreement. The recordings were deleted after transcription.

## Analysis

Data were analysed according to Hsieh and Shannon's conventional content analysis, which is appropriate when the aim of the study is to describe an under- or unexplored phenomenon [24]. The analysis started with reading the transcripts repeatedly to obtain a sense of the whole (all authors). Then the transcripts were read word by word, and exact words that appeared to capture key thoughts or concepts were highlighted (two of the authors). Notes were made to catch the first impressions, thoughts and initial analysis, and subsequently codes were derived based on more than one key thought (three of the authors). During the analysis, the transcripts were included in a table. Key words were marked yellow. Labels for codes were then transferred to the next column (initial coding scheme), and categories placed in the next column. All authors (of which two were experienced qualitative researchers) read the transcripts, and the preliminary derived codes were discussed untill consensus was reached. The codes were then sorted into different categories depending on how they were related and linked (three of the authors). The development of categories were also discussed among all authors untill consensus was reached. This was an iterative process, moving back and forth from transcripts to codes to categories.

## Ethical consideration

The study was approved by the Norwegian Center for Research Data (NSD) (Ref. no.951914), and was conducted in-line with recommendations in the Declaration of Helsinki [25]. The supervisors received oral and written information about the study purpose, and delivered signed written consents to participate. Due to the nature of a focus group, it was not possible to withdraw from the study. Participation was voluntary.

**Table 2. Participants' general characteristics.**

| Gender (n =) | |
|---|---|
| Male | 1 |
| Female | 33 |
| Educational background (n =) | |
| Registered nurse | 31 |
| Intellectual disability nurse | 3 |
| Age (years) | |
| Range | 23–58 |
| Mean | 39.5 |
| Experience (years) | |
| Range | 1–35 |
| In the current workplace, range | 0*-20 |
| Completed formal supervisor course (n =) | |
| Yes | 16 |

Abbreviations; Experience = as a registered nurse/intellectual disability nurse respectively.

* = 6 weeks.

## Results

A total of 34 RN/IDN student supervisors participated. Table 2 gives an overview of the participants' general characteristics.

The supervisors worked in psyciatric and substance abuse care, in home healthcare, in housing for disabled people, in a nursing home in primary healthcare, and in medical and surgical wards, including the children's ward in hospital.

Through the analysis we derived the category 'Shared responsibility for preparation' with subcategories a) Individual initiative, and b) University college facilitation. Illustrative quotes are marked with focus group (FG) number.

### Shared responsibility for preparation

All of the focus group participants found preparedness as essential for students to get the most out of their clinical placement, both personally and related to learning outcomes. For students to be prepared for clinical placement supervisors assumed a shared responsibility between the student and the university college. They expected the student to be acquainted with the patient groups as well as with possible learning outcomes for the specific clinical placement. Moreover, they expected that the university college provided the students with theoretical introduction and practical exercises in different practical procedures. These aspects were discussed and emphasized as important in all focus groups, across RN/IDN specialization and across primary- and hospital healthcare wards. In addition, there were no disagreements between participants within the focus groups regarding these aspects.

### Individual initiative

Throughout, there was an agreement in all focus groups that the student had an individual responsibility to be prepared, and hereby needed to show individual initiative. The supervisors in all focus groups emphasized that the students should reflect on their role as a student. Moreover, supervisors expected students to have insight into the learning outcomes related to the specific educational semester, as well as the potential learning opportunities at the specific

ward. Most focus groups (FG) also expressed increasing expectations of the students' preparedness from first to the last semester of the education program. One of the supervisors in FG 4 stated;

> «*That he thinks through what is expected as a student. That he knows what he is good at, and that he is prepared for those things is important for the third year.*» *(FG 4)*

This was supported by the other supervisors in the group. Supervisors in all groups also found it important that students had knowledge about different relevant diseases in the actual placement, both regarding pathology, medical treatment and nursing care needed in the specific group of patients. As stated in FG 6:

> «*That they are prepared and have knowledge about the patient group they will meet.*» *(FG 6)*

Some of the wards had developed an information letter about the placement that students got access to before the placement period. Here, supervisors expected the students to have read the information, even though this was not always the case. The supervisors in many of the focus groups also would like the students to visit the ward in advance.

In some of the focus groups, supervisors expressed that it was an advantage when the students had work experience from health or social services before starting their education and also that they worked during their studies. This was particularly emphasized as an advantage by IDN supervisors prior to clinical placements in substance abuse care and mental health work;

> «*To have som psychiatric experience or know a little of relationships, man, wholeness.*» *(FG 9)*

## University college facilitation

In all focus groups, supervisors expressed that they assumed that the university college is responsible for facilitating for students to access both theoretical and practical knowledge. They all expected that the students had been introduced to, and given the opportunity to perform different practical skills and procedures at least once at the university college. Hygienic principles and aseptic procedures were examplified as knowledge that could be transferred to many procedures, and the supervisors assumed that the university college enabled students to do so. Moreover, they pointed out that by letting the students practice on each other, one might perhaps lessen the discomfort one can feel when the procedures are to be performed on a patient for the first time. In FG 3 this was prompted:

> «*That they experience what it is like to have your teeth brushed by someone else, or be fed or having a bed bath.*» *(FG 3)*

and further

> «*Then the training can be done during the clinical placement, you get to try it several times, like injections, or, yes, wound care*» *(FG 3)*

The supervisors also expected that the university prepared the students for various clinical placements. For example, before clinical placement in surgical wards the participants expected the students to have knowledge of preoperative and postoperative nursing. A supervisor in FG 10 said:

«. . . then I expect that they have had lectures in pre and postoperative nursing. They should know what to look for when someone has just had surgery. What is the postoperative nursing, elimination, pain. . ...» (FG 10)

Prior to clinical placement in mental health care, the IDN supervisors expected the students to have knowledge of e.g. guardianship, law of coercion and force, and also of the most common drugs used in mental health treatment. FG 7 agreed on this statement:

«What is psychiatry? It is no longer just the medication. If they do not want to take their medicines, we can not force them, and then we have to do something else, and what do we do then?» (FG 7)

In several of the focus groups, the supervisors claimed that the university college had to prepare the students to be creative and adapt the nursing care to the current situation in the different care settings. This was esspecially emphasized when the students were in clinical placements in the patient's home in home health care services. In FG 4 this was concretized:

«Because we can enter a home and there is a man with a cardiac arrest in bed, what do you do?» (FG 4)

## Discussion

Our findings show that supervisors had expectations regarding students' individual initiative to be prepared for clinical placement, as well as to the university college facilitation of such preparedness. The focus groups consisted of participants from various primary healthcare wards, as well as hospital wards, and both RNs and IDNs, still there were no disagreements on these issues.

Supervisors perceived that students should take their part of the responsibility for being prepared for clinical placements. Moreover, they emhasized that this would require individual initiative, since the students would have to reflect, search for relevant literature regarding specific conditions or even visit the actual ward prior to the placement period. This is in-line with findings from e.g. Chipchase et al [18], who emhasized that students should accept a share of the responsibility for planning and preparing for the learning experience. Webb et al. [26] also found that supervisors wants students who are well prepared theoretically in front of clinical placements. In a recent study, supervisors rated professionalism and willingness as the most important characteristics in nursing students, followed by personal attributes. The authors concluded that further strengthening learning opportunities related to these characteristics in the nursing curriculum may improve the students' preparedness for clinical learning [27].

The supervisors also discussed the need to combine theoretical knowledge and practical skills, to be able to meet patients as whole human beings. According to Bloom [28] there are three domains of learning: 1. Knowledge (cognitive, mental skills), 2. Skills (psychomotor, manual or physical skills), and 3. Attitude (affective, growth in feelings). The clinical environment is a place for students to practice and gain both confidence and competence in all three of these domains [29]. Clinical training is regarded as an integral part of learning and education in nursing [5, 30]. Furthermore, the clinical learning environment plays an important role in turning nursing students into professionals and preparing them to function as nurses [31]. Nevertheless, supervisors expected that students had both knowledge and skills *before* their clinical placement.

The supervisors expected the university college to provide students with both theoretical knowledge and practical training relevant for the specific clinical placement. In a recent study across primary and tertiary healthcare in the same county, we found that RNs/IDNs perceived that students should learn technical skills in the university college first, and then get more training in clinical placement [32]. This was supported by findings from the focus group discussions in the current study.

Earlier studies have emphasized that supervising nursing students affects RNs' daily work to different degrees, depending on students' knowledge, progress and willingness to learn, but also on the supervisors themselves [33, 34]. Moreover, studies indicate that the nursing students are not psychologically prepared for clinical placements [35, 36]. In the current study, issues related to the supervisors themselves were not discussed. The gap between supervisors' expectations, and students' preparedness could lead to challenges both regarding the establishment of a constructive supervisor-supervisee relationship, the supervisors' experience of their role as a burden, and to the students' clinical learning. According to Kalyani et al. [37], educational policymakers should focus on increasing supervisors' influence on management, education and clinical education, as well as teaching positive and constructive strategies to cope with inadequate educational contexts. Donough et al. [38] also suggest a need for continuous professional development for clinical supervisors by means of in-service training.

Our findings indicate that RN/IDN supervisors expect for students to show individual initiative to get prepared for clinical placements. In addition, they assume that the university college has ensured students' preparedness related to theoretical knowledge and procedural competence. Even though the importance of supervisors' in students' clinical placements has been emphasized in many studies [5–16], participants in our study did not discuss their own role in student preparation.

## Limitations

Within the qualitative research study lies the lack of generalizability of study findings. Moreover, the study presented here has some further limitations. First, the data was collected by seven different interviewers., which may have introduced a researcher-bias. Still, they all had experience in conducting interviews, and had long experience from both clinical practice as RNs/IDNs and educators. In addition, to minimize the possible differences, the interview guide was discussed in advance among the researchers to reach a joint understanding of the questions and how to perform the focus group interviews.

Second, in this study, both RNs and IDNs were included, but few IDNs participated. IDNs are authorized healthcare personnel and work in different clinical settings. Inclusion of more IDNs might have proved additional data. Even if they were few in numbers, they contributed with important experiences from being supervisors. In addition, several of the supervisors who were RNs also supervised IDN students, adding to the IDN student perspective.

Credibility was ensured through several and iterative analysis and discussions both between the authors and with the research group members. The research group consisted of 12 IDNs and RNs from both clinical wards and the university college. The supervisors represented both RNs and IDNs, from a variety of clinical wards both in primary- and hospital healthcare, indicating transferability of findings. To ensure dependability, the first author verified the authenticity of the recordings of the interviews against the transcripts. Unfortunately, non-verbal expressions during the interviews were not charted and included in the analysis. This may have had an impact on the results of our study.

## Conclusion

Findings indicate that the supervisors across primary and tertiary healthcare have expectations to students' individual initiative to get prepared for clinical placement, as well as the

educational institutions' fascilitation thereof. In contrast, they did not discuss their own role in getting students prepared for clinical placements, or how they could support students at the beginning of the clinical placement periode. Our findings indicate that there is a gap between supervisors expectations and reality, which could impact both their relationship to the student, as well as the students' learning process. Clinical placement quality improvement initatives should focus on closing this gap.

## Relevance to clinical practice

Further studies should focus on how supervisors could contribute to students' preparedness, together with the educational institutions. Moreover, students' and supervisors' perspectives should be compared and efforts made to align these, aiming for expectations to be in-line with reality. Educational policymakers should focus on increasing supervisors' influence on clinical placements, as well as the importance of collaboration between the supervisors and employees in the educational institutions. In addition, supervisors should be given the opportunity to access formal competence in supervision.

## Supporting information

**S1 File.**
(DOCX)

## Acknowledgments

Inger Hjelmeland, Ina Kristin Blågestad, Anne Grethe Gregersen, Lars Gunheim-Hatland, Anne Herwander Kvarsnes, Mona Martinsen, Richard Olsen, and Wenche Charlotte Hansen are acknowledged for participating in the planning of the study, as well as in development of the interview guide. In addition, Anne Grethe Gregersen and Lars Gunheim-Hatland are acknowledged for participating in the focus group interviews.

## Author Contributions

**Conceptualization:** Ann-Chatrin L. Leonardsen, Siri E. Brynhildsen, Mette T. Hansen, Vigdis A. Grøndahl.

**Data curation:** Ann-Chatrin L. Leonardsen, Siri E. Brynhildsen, Mette T. Hansen, Vigdis A. Grøndahl.

**Formal analysis:** Ann-Chatrin L. Leonardsen, Siri E. Brynhildsen, Mette T. Hansen, Vigdis A. Grøndahl.

**Funding acquisition:** Ann-Chatrin L. Leonardsen.

**Investigation:** Ann-Chatrin L. Leonardsen, Siri E. Brynhildsen.

**Methodology:** Ann-Chatrin L. Leonardsen, Siri E. Brynhildsen, Vigdis A. Grøndahl.

**Project administration:** Ann-Chatrin L. Leonardsen.

**Resources:** Ann-Chatrin L. Leonardsen.

**Writing – original draft:** Ann-Chatrin L. Leonardsen.

**Writing – review & editing:** Ann-Chatrin L. Leonardsen, Siri E. Brynhildsen, Mette T. Hansen, Vigdis A. Grøndahl.

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
