## [Decision Letter · Decision Letter 0]

22 Mar 2021

PONE-D-21-03784

Nursing supervisors’ perspectives on student preparedness before clinical placements- a focus group study

PLOS ONE

Dear Dr. Leonardsen,

Thank you for submitting your manuscript to PLOS ONE. After careful consideration, we feel that it has merit but does not fully meet PLOS ONE’s publication criteria as it currently stands. Therefore, we invite you to submit a revised version of the manuscript that addresses the points raised during the review process.

We look forward to receiving your revised manuscript.

Kind regards,

Prof, Mojtaba Vaismoradi, PhD, MScN, BScN

Academic Editor

PLOS ONE

Journal Requirements:

2. Please include a copy of the interview guide used in the study, in both the original language and English, as Supporting Information, or include a citation if it has been published previously.

4. Thank you for submitting the above manuscript to PLOS ONE. During our internal evaluation of the manuscript, we found significant text overlap between your submission and the following previously published works, some of which you are an author.

https://bmcmededuc.biomedcentral.com/track/pdf/10.1186/1472-6920-12-112.pdf

https://link.springer.com/article/10.1186/s12909-017-0966-4?code=3b519ef9-c50c-49a5-94ee-5ab75637dc44&error=cookies_not_supported

Please revise the manuscript to rephrase the duplicated text, cite your sources, and provide details as to how the current manuscript advances on previous work. Please note that further consideration is dependent on the submission of a manuscript that addresses these concerns about the overlap in text with published work.

Reviewers' comments:

Reviewer #1: Thank you so much for the opportunity to review your paper. First of all, this topic is very practical and the information is meaningful to the nurse supervisors, educators, and nurse students as well. Following recommendations may help improve the paper quality:

1. This paper did not indicate that the observations were charted during the interview and therefore, they were not included in the analysis. For interviews, verbal expressions are important, but the non-verbal information is equally important to the study. Thus, the results might be affected when the non-verbal information is missed.

2. Some spelling errors: such as Ln 102, "purposeful", it is better to use the word " purposive". Ln 301, Fidnings-findings.

Reviewer #2: This is a qualitative study aims at exploring supervisors’ perspectives on student preparedness for clinical placements.

The manuscript is well-written and explains the research clearly. The findings provide useful information for the readers.

I enjoyed reading this and I have one minor comment:

1. sample (lines 102 -109): i would suggest not to use first person such as "we"

Reviewer #3: I was very interested in the topic of focus group interviews on supervisors' perspectives of students in practice. From my experience managing practitioners, I don't think the results of this study's interviews are new to the nurses currently working in hospitals. However, when it is read by nursing students or professors leading students, it seems to be helpful to some extent.

The overall regret about the result is that the concept derived from the interview that lasted for 5 months is small. Interviews must have been conducted with 34 people for 5 months, but there is no information on how many interviews the data was saturated. I wonder if the concept has been sufficiently drawn from the interview data. Although it has been suggested as a limitation of the study, it is regrettable that there is no mention of IDN. Thank you.

Reviewer #4: 1. The paper has grammatical and needs grammar editing to improve its quality

2. The introduction needs to be improved to bring out the problem

3. The paper should also highlight the policy implications of the findings.

4. What do the authors mean by ‘piloting within the research group’? Sounds confusing to the reader.

5. How did the interviewers relate to the participants? What was their positioning within the study?

Reviewer #5: The authors have explored an area which required research. Findings are genuine. It would have been really nice if results were more clearly depicted. Conclusions are sound and upto the mark. Kindly publish, if possible.

Reviewer #6: Thanks for giving me the opportunity to review the manuscript

It is a good study and authors have selected an important topic. However there is a scope to improve this manuscript and some of my suggestions are given below:

The introduction needs to provide a better background on the existing opportunities for nursing students preparedness before clinical placement and the role of nursing supervisors in the selected setting in order for larger global readers to understand the context and need of the study.

Method section:

Total population of nursing supervisors in the selected setting needs to be mentioned.

Mention the sample size in the sample section and the basis on which sample size is calculated for the study also needs to be justified .

Line no 110-111 not very clear what do you mean by saying that they were asked to participate by their manager? Since the cadre and nomenclature related to different nursing posts differ worldwide so clearly mention who is a manager in these health care setting

Although you have mentioned some of the questions of the semi structured interview guide but it would be good if you attach interview guide as a supplementary file.

Mention and justify how many focus group discussions were done.

Write few lines to summarize the result and the discussion section at the end

Add the recommendations and also comment on the generalizability of the study results.

---

## [Author Response · Author response to Decision Letter 0]

6 Apr 2021

PLOS ONEE

Prof, Mojtaba Vaismoradi, PhD, MScN, BScN

Academic Editor

Re-submission of manuscript: PONE-D-21-03784

Nursing supervisors’ perspectives on student preparedness before clinical placements- a focus group study

Dear Editor 

We would like to thank the editors and reviewers for their comments on our manuscript. Below You will find the issues raised, followed by our response to these remarks. 

Comments from the editor 

Comment 1: a) If there are ethical or legal restrictions on sharing a de-identified data set, please explain them in detail (e.g., data contain potentially identifying or sensitive patient information) and who has imposed them (e.g., an ethics committee). Please also provide contact information for a data access committee, ethics committee, or other institutional body to which data requests may be sent. b) If there are no restrictions, please upload the minimal anonymized data set necessary to replicate your study findings as either Supporting Information files or to a stable, public repository and provide us with the relevant URLs, DOIs, or accession numbers. Please see http://www.bmj.com/content/340/bmj.c181.long for guidelines on how to de-identify and prepare clinical data for publication. For a list of acceptable repositories, please see http://journals.plos.org/plosone/s/data-availability#loc-recommended-repositories.

Response: When including transcript from the focus group interviews, there is a risk that the discussions themselves, or references to actions, patients, places or challenges, may identify specific wards/locations and hereby patients. Therefore, we did not include this as supporting information. Moreover, the transcripts of the interviews are in Norwegian, and therefore not accessible to an international audience. Hence, we wish to keep our statement- if possible. 

Comment: 4. Thank you for submitting the above manuscript to PLOS ONE. During our internal evaluation of the manuscript, we found significant text overlap between your submission and the following previously published works, some of which you are an author.

https://bmcmededuc.biomedcentral.com/track/pdf/10.1186/1472-6920-12-112.pdf

https://link.springer.com/article/10.1186/s12909-017-0966-4?code=3b519ef9-c50c-49a5-94ee-5ab75637dc44&error=cookies_not_supported

Please revise the manuscript to rephrase the duplicated text, cite your sources, and provide details as to how the current manuscript advances on previous work. Please note that further consideration is dependent on the submission of a manuscript that addresses these concerns about the overlap in text with published work.

Response: We have revised the manuscript thoroughly, and hope that our revisions now are deemed not-overlapping with present publications. 

Reviewers' comments

Reviewer #1

Comment 1: Thank you so much for the opportunity to review your paper. First of all, this topic is very practical and the information is meaningful to the nurse supervisors, educators, and nurse students as well. Following recommendations may help improve the paper quality:

1. This paper did not indicate that the observations were charted during the interview and therefore, they were not included in the analysis. For interviews, verbal expressions are important, but the non-verbal information is equally important to the study. Thus, the results might be affected when the non-verbal information is missed.

Response: We appreciate this input. Due to several researchers conducing the interview, as well as the large number of interviews, non-verbal information was not included in the analysis. We have added this as a further limitation to the study. Please see manuscript with track changes. 

Comment 2: 2. Some spelling errors: such as Ln 102, "purposeful", it is better to use the word " purposive". Ln 301, Fidnings-findings.

Response: We thank the reviwer for this comment, and have consequently revised the manuscript accordingly. Please see manuscript with track changes. 

Reviewer #2

Comment 1: This is a qualitative study aims at exploring supervisors’ perspectives on student preparedness for clinical placements.

The manuscript is well-written and explains the research clearly. The findings provide useful information for the readers. I enjoyed reading this and I have one minor comment:

1. sample (lines 102 -109): i would suggest not to use first person such as "we"

Response: We thank the reviewer for positive inputs. The first person has been deleted in the revised manuscript. 

Reviewer #3

Comment 1: I was very interested in the topic of focus group interviews on supervisors' perspectives of students in practice. From my experience managing practitioners, I don't think the results of this study's interviews are new to the nurses currently working in hospitals. However, when it is read by nursing students or professors leading students, it seems to be helpful to some extent.

The overall regret about the result is that the concept derived from the interview that lasted for 5 months is small. Interviews must have been conducted with 34 people for 5 months, but there is no information on how many interviews the data was saturated. I wonder if the concept has been sufficiently drawn from the interview data.

Response: We thank the reviewer for this input. Saturation is a discussed term in qualitative research, and there is no consensus of whether it is appropriate to refer to. Our experience was that some themes were repeated across interviews, but repeated interviews also added new information/themes. Hence, we can not claim that saturation has been reached. 

Comment 2: Although it has been suggested as a limitation of the study, it is regrettable that there is no mention of IDN. Thank you.

Response: We are also sorry that we were not able to recruit more IDNs to this study. Some places in the results section, we specify that IDNs also related to statements or discussions. In the revised manuscript, we have included some more information about IDNs’ statements. Unfortunately, research on IDNs’ supervision or practice is lacking, so we have not been able to identify studies to include in the discussion section. 

Reviewer #4: 

Comment 1: The paper has grammatical and needs grammar editing to improve its quality

Response: We have revised the manuscript for grammatical errors. Please see manuscript with track changes. 

Comment 2: The introduction needs to be improved to bring out the problem

Response: We have tried to highlight «the problem» further- please see manuscript with track changes. 

Comment 3: The paper should also highlight the policy implications of the findings.

Response: In the revised manuscript we have added policy implications, under the «Relevance to clinical practice» section. Please see manuscript with track changes.

Comment 4: What do the authors mean by ‘piloting within the research group’? Sounds confusing to the reader.

Response: The study was initiated in a research group consisting of a total of 12 participants. The interview guide was piloted in this group. In the revised manuscript, we have tried to clarify this. 

Comment 5: How did the interviewers relate to the participants? What was their positioning within the study?

Response: The interviewers were not familiar to the participants. The interviewers were participants in the research group initiating the study. We have tried to clarify this in the revised manuscript. Please see manuscript with track changes. 

Reviewer #5

Comment:The authors have explored an area which required research. Findings are genuine. It would have been really nice if results were more clearly depicted. Conclusions are sound and upto the mark. Kindly publish, if possible.

Response: We thank the reviewer for appreciating our study. 

Reviewer #6

Comment 1: Thanks for giving me the opportunity to review the manuscript

It is a good study and authors have selected an important topic. However there is a scope to improve this manuscript and some of my suggestions are given below:

The introduction needs to provide a better background on the existing opportunities for nursing students preparedness before clinical placement and the role of nursing supervisors in the selected setting in order for larger global readers to understand the context and need of the study.

Response: We thank the reviewer for this input, and have revised the background. Please see manuscript with track changes. 

Comment 2: Method section:

Total population of nursing supervisors in the selected setting needs to be mentioned.

Response: Unfortunately, we do not have any overview of the number of supervisors in the selected setting. The RNs/IDNs circulate the supervisory role, and there are no statistics presenting the number of employed healthcare personnel. In fact, this is a national problem, stated by the gouvernment.

Comment 3: Mention the sample size in the sample section and the basis on which sample size is calculated for the study also needs to be justified .

Response: We are not familiar with sample size calculations in qualitative studies. Hence, we did not perform any sample size calculations. 

Comment 4: Line no 110-111 not very clear what do you mean by saying that they were asked to participate by their manager? Since the cadre and nomenclature related to different nursing posts differ worldwide so clearly mention who is a manager in these health care setting

Response: We thank the reviewer for pointing this out. We have revised the methods section, and we hope that this now is more clear. Please see manuscript with track changes. 

Comment 5: Although you have mentioned some of the questions of the semi structured interview guide but it would be good if you attach interview guide as a supplementary file.

Response: The interview guide has been attached as a supplementary file. 

Comment 6: Mention and justify how many focus group discussions were done.

Response: We chose the number of groups pragmatically, as stated in the manuscript. In qualitative research environments, there are no consensus on the number of participants in interviews, or the number of focus groups in such discussions. Hence, we aimed at including participants from two surgical and two medical hospital wards, and a variety of primary healthcare wards (n=4). 

Comment 7: Write few lines to summarize the result and the discussion section at the end

Response: The results section has been summarized in the beginning of the discussion section, We have added a summary of the discussion section as recommended. 

Comment 8: Add the recommendations and also comment on the generalizability of the study results.

Response: We have added recommendations, and have commented on the generalizability of the study results. Please see manuscript with track changes. 

Concluding remarks

The comments provided here are submitted in adition to a manuscript with track changes, as well as a clean copy. We hope our revisions are deemed sufficient, and that the paper will be accepted for publication in PLOS One. Thank You for considering our revised manuscript for publication.

Yours sincerely

Ann-Chatrin Linqvist Leonardsen

---

## [Decision Letter · Decision Letter 1]

17 May 2021

Nursing supervisors’ perspectives on student preparedness before clinical placements- a focus group study

PONE-D-21-03784R1

Dear Dr. Leonardsen,

We’re pleased to inform you that your manuscript has been judged scientifically suitable for publication and will be formally accepted for publication once it meets all outstanding technical requirements.

Kind regards,

Prof Mojtaba Vaismoradi, PhD, MScN, BScN

Academic Editor

PLOS ONE, 

& Nord Universitet, Norge

---

## [Editor Report · Acceptance letter]

19 May 2021

PONE-D-21-03784R1 

Nursing supervisors’ perspectives on student preparedness before clinical placements- a focus group study 

Dear Dr. Leonardsen:

I'm pleased to inform you that your manuscript has been deemed suitable for publication in PLOS ONE. Congratulations! Your manuscript is now with our production department. 

Kind regards, 

on behalf of

Professor Mojtaba Vaismoradi 

Academic Editor

PLOS ONE